

**Load-resistance analysis: An alternative approach to tsunami damage assessment applied**
**to the 2011 Great East Japan tsunami**
Anawat Suppasri[1], Kwanchai Pakoksung[1], Ingrid Charvet[2], Constance Ting Chua[3], Noriyuki
Takahashi[4], Teraphan Ornthammarath[5], Panon Latcharote[6], Natt Leelawat[7] and Fumihiko
Imamura[1]
[1]International Research Institute of Disaster Science, Tohoku University
(468-1 Aramaki-aza Aoba, Aoba-ku, Sendai 980-0845, Japan)
[2]Department of Statistical Science, University College London, United Kingdom
(Gower Street, London, WC1E 6BT)
[3]Asian School of the Environment, Nanyang Technological University
(N2-01C-39, 50 Nanyang Avenue, Singapore 639798)
[4]Department of Architecture and Building Science, School of Engineering, Tohoku University
(6-6-11-1 Aramaki-aza Aoba, Aoba-ku, Sendai 980-8579, Japan)
[5]Department of Civil and Environmental Engineering, Faculty of Engineering, Mahidol University
(25/25 Puttamonthon, Nakorn Pathom, 73170, Thailand)
[6]Department of Sustainable Development Technology, Faculty of Science and Technology, Thammasat
University
(99 Moo 18, Phaholyothin Road, Tambon Klong Nung, Amphoe Klong Luang, Pathum Thani 12120,
Thailand)
[7]Disaster and Risk Management Information Systems Research Group, Department of Industrial
Engineering, Faculty of Engineering, Chulalongkorn University
(Phayathai Road, Pathumwan, Bangkok 10330 Thailand)

**Abstract**
Tsunami fragility functions describe the probability of structural damage to tsunami flow characteristics.
Fragility functions developed from past tsunami events (e.g. 2004 Indian Ocean tsunami) are often
applied directly, without modifications, to other areas at risk of tsunami for the purpose of damage and
loss estimations. Consequentially, estimates carry uncertainty due to disparities in construction
standards and coastal morphology between the specific region for which the fragility functions were
originally derived and the region where they were being used. The main objective of this study is to
provide an alternative approach to assessing tsunami damage, especially for buildings in regions where
previously developed fragility functions do not exist. A damage assessment model is proposed in this
study, where load-resistance analysis is performed for each building by evaluating hydrodynamic forces,
buoyancies and debris impacts and comparing them to the resistance forces of each building. Numerical
simulation was performed in this study to reproduce the 2011 Great East Japan tsunami in Ishinomaki
city, which is chosen as a study site. Flow depths and velocities were calculated for approximately 20,
000 wooden buildings in Ishinomaki city. Similarly, resistance forces (lateral and vertical) are estimated
for each of these buildings. The buildings are then evaluated for its potential to collapse. Results from
this study reflect a higher accuracy in predicting building collapse when using the proposed load-
resistance analysis as compared to previously developed fragility functions in the same study area.
Damage is also observed to have likely occurred before flow depth and velocity reach maximum values.
With the above considerations, the proposed damage model might well be an alternative for building
damage assessments in areas which have yet to be affected by modern tsunami events.

**Figures with higher quality can be seen in supplemental file.**






## 1. Introduction

The 2011 Great East Japan earthquake generated a large tsunami which damaged and destroyed more than 250, 000 buildings (MLIT, 2012). Building damage characteristics from the 2011 event have since been well-studied and in most cases, used to develop tsunami damage fragility functions (Suppasri et al., 2015). Tsunami damage fragility functions describe the probability of structural damage to tsunami flow characteristics, i.e. flow depth, flow velocity and hydrodynamic force. Fragility functions have been developed from past events (e.g. 2004 Indian Ocean, 2010 Chile and 2011 Great East Japan tsunamis) and are often applied directly, without modifications, to other areas facing tsunami risk for damage and loss assessments (Suppasri et al., 2016). The resulting damage estimates carry uncertainty related to differences in construction standards and coastal morphology between the specific region for which the fragility functions were originally derived and the region where they are being used.

Tsunami fragility functions are modelled using tsunami flow characteristics and building damage information. These characteristics are usually obtained through either: – 1) Field survey and 2) Combined analyses from tsunami numerical simulation and satellite imagery (Charvet et al, 2017). In a field survey, maximum flow depth measured from tsunami water traces are typically used as explanatory variables of damage. Building damage data is obtained from on-site observations (Reese et al., 2007, Valencia et al., 2011, Suppasri et al., 2015 and Triantafyllou et al., 2018). This method combines tsunami numerical simulation, which generates maximum flow depth, flow velocity and hydrodynamic force values, and satellite image analysis of pre- and post- disaster images to obtain building damage data (Koshimura et al., 2009, Omira et al., 2010 and Suppasri et al., 2011) or other damage data set such as damaged marine vessels (Suppasri et al., 2014), damaged bridges (Shoji and Nakamura, 2017) as well as aquaculture rafts and eelgrass (Suppasri et al., 2018). In both methods, only maximum values of tsunami flow characteristics are considered as explanatory variables. However, a given damage level might have already occurred before these maximum values are attained, thus building damage may be underestimated. Additionally, recent studies have shown tsunami hydrodynamic force to be an important explanatory parameter (Macabuag et al., 2016), flow velocity at time of occurrence (Song et al., 2018) and floating debris (Macabuag et al., 2018) are all factors when assessing building damage.

In order to obtain fragility functions for areas where tsunami data is not yet available, it is necessary to model the deterministic processes relating tsunami characteristics to the capacity of the structure to resist resulting loads. This allows for the structural characteristics information specific to the buildings of a region to be taken into account, as well as bypassing the use of potentially biased observed values for the explanatory variables. This study investigates interactions between tsunami loading and the resistance of a system (in this case the resistance of a building) through an analytical model to infer tsunami damage. The objective is to provide an alternative approach to assessing tsunami damage especially for buildings in areas where previously developed fragility functions do not exist. As part of this study, tsunami characteristics at the time of damage occurrence will be investigated and used in the proposed model to provide a complementary insight into the relationship between structural damage and tsunami flow characteristics.

The analytical model is defined following an overview of tsunami flow characteristics and their effects on buildings. Next, the study site and building damage data set used to demonstrate the application of the model are presented. Two major components of the model are then discussed: tsunami numerical simulation and the estimation of resisting forces. Model results are compared to other building damage assessment estimates and observations in order to examine their applicability in building damage estimation. In addition, because structural damage is usually presented in a qualitative manner, most tsunami damage assessments may not be readily usable by private or governmental organisations. Therefore, a financial metric converting existing structural damage levels into financial cost ratios is proposed.



98

## 2. Alternative approach to tsunami damage assessment

Damage by tsunamis to infrastructure are caused by many factors such as tsunami forces, impact of waterborne debris, building characteristics and scouring of foundations (Kelman and Spence, 2004). Forces generated by a tsunami can be estimated by classifying them according to their flow conditions and characteristics. Hydrodynamic force is generated by the pressure from flowing waters around the structure, and is influenced by flow velocity, depth and density of the water as well as the geometry and angle at which the tsunami hits the structure (Nadal et al., 2010). When hydrodynamic force is used in tsunami science, it usually refers to the drag force which is directly proportional to the square of flow velocity. Debris impact force is driven by tsunami flow. Tsunami-borne debris, while not a direct action of tsunami flow, can cause substantial damage to buildings. It can result in the reduction of load-bearing capacity in a building, and therefore the reduction in structural resistance to lateral loads and buoyancy forces (Nadal et al., 2010).

The approach taken in this study is an adaptation from Latcherote et al (2017) where they analysed and compared the overturning mechanism with resisting moment for six overturned reinforced concrete buildings in Onagawa town. Similarly, the proposed damage model performs load-resistance analysis for each building by evaluating hydrodynamic forces, buoyancy forces and debris impacts and comparing them to the resistance of each building. There are two general types of resistance that a building provides. First, it provides lateral resistance which is designed to counter loads that are perpendicular to and imposed on walls. Second, the weight of the buildings acts as downward-acting (vertical) resistance against buoyancy forces or upward-acting loads from wind and seismic activities. The resistance force from pile foundation was also one of the components examined Latcherote et al. (2016). However, because wooden buildings were used for this study, the resistance force from pile foundation was not considered.

Global stability failure in a building can be a result of either sliding or overturning as a solitary body, often with minimal damage to structural/non-structural components (Yeh et al, 2014). Overturning refers to the rotation of a building about its foundation where it has failed. Sliding, on the other hand, is the horizontal translation of a building from its original position (Yeh et al, 2014). The two mechanisms are modelled separately in this study to determine the predominant mechanism for building collapse. Differences in the forces and resistance involved in these mechanisms were considered when performing load-resistance analysis:

(1) Sliding/Non-submerged at the point of impact (**Fig. 1 (a)**): Only horizontal hydrodynamic force, debris impact and lateral resistance of the building were considered in this case. A building collapses if the compounded hydrodynamic and debris impact forces are greater than the lateral resistance of the building.

(2) Overturning/Submerged (**Fig. 1 – (b)**): A building collapses when the overturning moment from hydrodynamic and buoyancy forces is greater than the resisting moment from the building weight. Under such circumstances, the building can either be fully submerged as illustrated in **Fig. 1 (b)** or surrounded by water with no water inside. In the former case, when the building is completely inundated, forces from the exterior of the building are cancelled out. The latter is the worst-case scenario and is assumed for subsequent analyses of overturning mechanisms in this study.



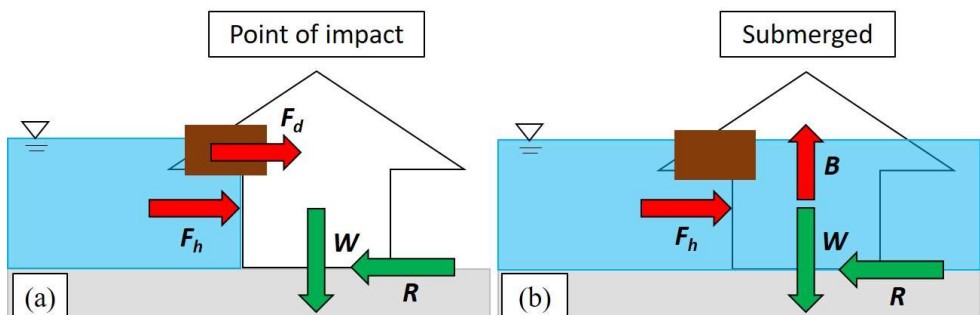

**Fig.1** Two failure mechanism considered in this study: (a) Sliding and (b) overturning.

**2.1 Selection of study site**
There were many possible areas for studying building damage from the 2011 Great East Japan tsunami
event. A suitable study site needs to be highly representative of the processes being modelled, without
excessive contributions of un-modelled effects. In addition, a previously investigated area would allow
for a fair assessment of the analytical model's results. Ishinomaki City, Miyagi Prefecture was therefore
selected as the area displayed the following characteristics:
1. Less impact from wave amplification: Ishinomaki City is located on a plain coast which reduces
the effects of wave amplification unlike coastal towns located along the Sanriku Ria Coast
2. Less impact from floating debris: The populated areas of Ishinomaki are far from fishing ports and
storage facilities, many of which were damaged by the tsunami and generated floating debris, which
can magnify building damage. Floating debris from broken pine trees can also be excluded from
consideration as the coastal pine forest along the city survived.
3. Less impact from wave directions: The effects from varying wave directions are minor as most of
the buildings were lined facing the shoreline and the direction of wave attack was perpendicular to
the front of the buildings.
4. Largest sample size: The number of buildings affected by the 2011 event was largest in
Ishinomaki City amongst cities along the plain coast.
5. Previously developed fragility functions: Fragility functions have been previously developed for
the populated areas of Ishinomaki City (Charvet et al., 2014). A new study from Hasegawa et al.,
(2018) provides an excellent opportunity to compare the proposed method in this study with the
established model.

**2.2 Building damage data**
Detailed building damage data from field observations was obtained from the Ministry of Land,
Infrastructure and Transportation and Tourism (MLIT) (MLIT, 2012) (**Fig. 2**) to test the applicability
of the proposed building damage model. The data consists of building size (length and width), number
of stories, construction material and interpolated measured maximum flow depth of each building. Each
building was also classified according to their observed damage. There are a total of six damage levels
in the classification scheme by MLIT. Low damage levels (i.e. levels 1 – 4) are easily misclassified in
damage assessments due to overlapping descriptions in the classification scheme (Leelawat et al., 2014),
whereas damage levels 5 and 6 are straightforward in their definitions (**Fig. 3**). "Washed away" and
"destroyed" (levels 5 and 6) refer to structures which are irreparable. In this study, the two levels
"washed away" and "destroyed" are considered since sliding and overturning mechanisms fall into the
aforementioned categories. As opposed to lower damage levels, these damage modes are driven by the
structural properties of these buildings, thus only buildings damaged at these levels were used for this
study. The building type considered in this pioneer study is only wooden residential houses due to their
large sample size in this area.



**Fig.2** (Left) Distributions of building types and (Right) building damage levels.

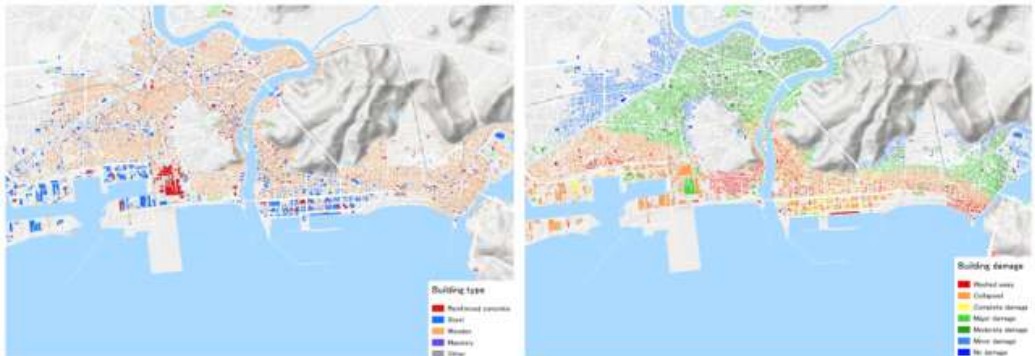

**Fig. 3** Building damage levels and collapsed condition considered in this study.

## 2.3 Numerical simulation of the 2011 tsunami and damage inducing forces

Tsunami flow characteristics (flow depth, velocity and hydrodynamic force) at the point of damage
occurrence were estimated in a time series analysis of the 2011 Great East Japan tsunami, which was
reproduced by numerical simulation. The numerical model computed tsunami propagation and run-up
by using a set of nonlinear shallow water equations which were solved by staggered leap-frog finite
difference scheme, and bottom frictional values were written using Manning's formula (Suppasri et al.,
2011, Charvet et al., 2015 and Macabaug et al., 2016). The model set-up includes the preparation of
bathymetry and topography data – a nested grid system consisting of six computational domains – 1215
m (Region 1), 405 m (Region 2), 135 m (Region3), 45 m (Region 4), 15 m (Region 5) and 5 m (Region
6) was used for the study area (**Fig. 4**). Only at the finest resolution (Region 6) were different Manning's
roughness coefficients specified according to land   use types and building density, as the effect of
bottom friction on tsunami propagation in deep waters negligible. Tidal level was set to tide conditions
at the time of tsunami occurrence, and simulation time was set to three hours. Initial water surface
elevation was assumed to follow sea floor deformation, the fault parameters proposed by Tohoku
University model (Imamura et al, 2016) were selected to reproduce the 2011 Great east Japan tsunami.
Results of numerical simulation are shown in **Fig. 5**.
The accuracy of model is validated by comparing measured tsunami trace heights and modelled results
(**Fig. 6**) using Aida's *K* and *κ* (Aida, 1978) as defined in equations (1) - (3) below.


$$\log K = \frac{1}{n} \sum_{i=1}^{n} \log K_i \qquad (1)$$

$$\log \kappa = \sqrt{\frac{1}{n} \sum_{i=1}^{n} (\log K_i)^2 - (\log K)^2} \qquad (2)$$

$$K_i = \frac{x_i}{y_i} \qquad (3)$$


Where, $x_i$ and $y_i$ are the measured and simulated tsunami trace heights (Mori et al., 2012) at point $i$.
Consequently, $K$ is regarded as a correction factor to adjust the modeled values to fit the actual tsunami
averaged over several locations; $\kappa$ is defined as a measure of the fluctuation or deviation in $K_i$. Values
of Aida's $K$ and $\kappa$ are 1.04 and 1.32 respectively. The corrected tsunami simulation produced tsunami
flow depths which are a close match to the measured tsunami trace heights and satisfy the guideline of
the Japan Society of Civil Engineers (JSCE) ($0.95 < K < 1.05$ and $\kappa < 1.45$) (JSCE, 2016). Hence,
tsunami flow depths and velocities in Ishinomaki City of higher accuracy were reproduced.

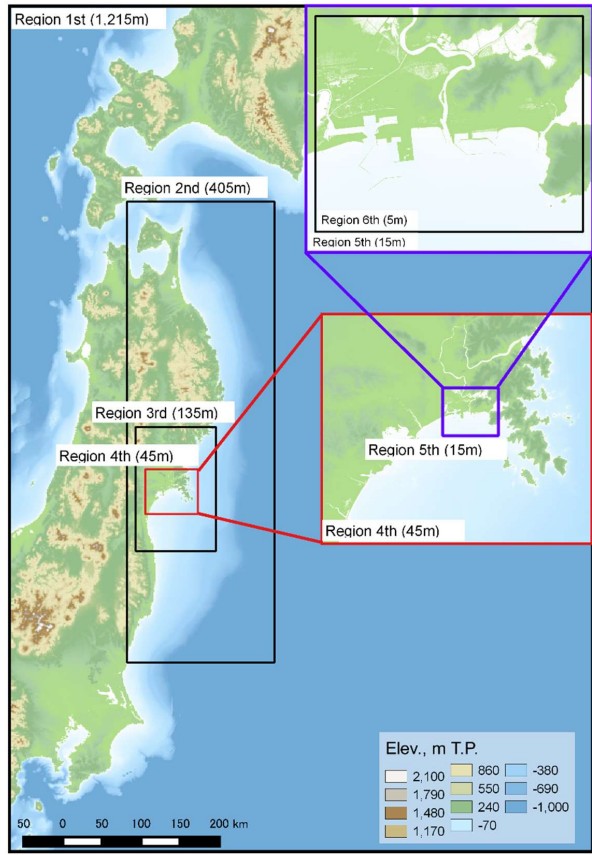

**Fig.4** Computational regions in this study.

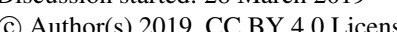


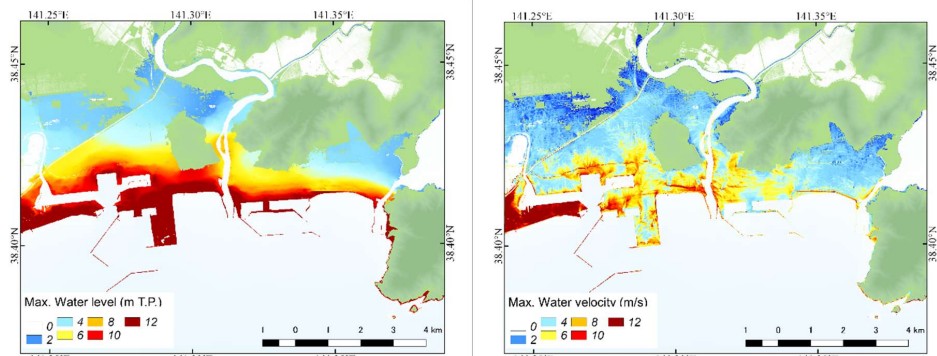

**Fig. 5** Results of tsunami numerical simulation: (Left) Maximum flow depth and (Right) Maximum
flow velocity.


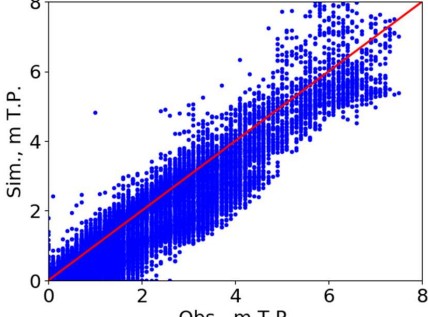

**Fig. 6** Validation of the simulated tsunami inundation heights using the observed tsunami trace
heights (Mori et al., 2012).


Results from the tsunami simulation were used to estimate tsunami-induced forces. Flow depth and
velocity values were captured at each time step of the simulation and at each building location for more
than 20,000 wooden buildings in Ishinomaki city. These values were then used to calculate
hydrodynamic force ($F_h$) though drag formula (equation (4)), debris impact force ($F_d$) through impulse-
momentum approach (equation (5)) as well as buoyancy force ($B$) (equation (6)) at each time step for
each building (**Fig. 1**).
$$F_h = \frac{1}{2} C_D \rho u^2 D \qquad (4)$$


$$F_d = m \frac{u}{\Delta t} \qquad (5)$$


$$B = \rho g V \qquad (6)$$


Where $C_D$ denotes the drag coefficient ($C_D = 1.5$), $\rho$ the density of water ($= 1,000$ kg/m³), $u$ the current
velocity (m/s), $D$ inundation depth (m), $m$ (kg) the weight of debris, $\Delta t$ the duration of impact ($= 0.7$ sec



for wooden wall), $g$ the gravitational acceleration and $V$ the submerged volume. This study follows the
recommended weights of floating debris by the American's Federal Emergency Management Agency
(FEMA, 2003) and Japan Society of Material Cycles and Waste Management (JSCWM, 2011), where
the estimates were approximately 500 kg for a pine tree, 3,000 kg for a vehicle, and buildings - 15,000
kg, 30,000 kg and 60,000 kg for moderately damaged, majorly damaged and collapsed buildings
respectively.

**2.4 Resistant forces**
In this study, the designed resistance of each building to withstand loads imposed on them is considered
as its damage threshold. One aim is to determine if the modelled tsunami induced forces (i.e.
hydrodynamic force, buoyancy force and debris impact force) for each building would exceed its
damage threshold and therefore, result in damage to the building. As mentioned earlier, differences in
the types of loads imposed and types of building resistance forces involved were considered when
modelling sliding and overturning mechanism of a building. Both mechanisms were modelled
separately. There are two types of resistant forces in a building i.e. vertical and lateral resistance. The
vertical resistance of a building is its weight, and in this study, it was assumed to be 3,000 kN/m² for
each building (Yokohama City, 2018). Vertical load-resistance analysis was used to determine
overturning mechanisms.
For the first time, lateral resistance ($R$) from the bearing wall of a building will be considered when
estimating building damage from tsunamis. The failure of lateral resistance of a building can imply that
sliding mechanisms are involved in its collapse. The bearing wall of a building must be able to resist
lateral loads imposed on them such as wind or seismic activity. The lateral resistance of each building
to earthquake and wind forces was calculated in accordance with Article 46 Enforcement Ordinance of
Building Standard Law (MLIT, 2018), and in which case, lateral resistance is the product of the lateral
strength of the bearing wall and the required wall length of each building. The lateral strength of the
bearing wall by Japanese housing design standard is 1.96 kN/m (MLIT, 2018).
Calculations for the required wall length would differ for both seismic and wind loads. Required wall
length for seismic loads can be derived by taking the building's floor area and multiplying it by its
design coefficient for seismic load (**Fig. 7**) as illustrated in Example 1. On the other hand, for wind
loads, the required wall length can be calculated by multiplying the design coefficients with the vertical
projection area (both the front and side of the building) as illustrated in Example 2. The vertical
projection area is the area defined by the building width or length multiplied by the floor height above
1.35 m (**Fig. 8**). As information on building heights in Ishinomaki city was not available at the point of
this study, 3.5 m, 2.7 and 2.1 m were assumed to be the heights of the first, second and third floors
respectively. Wooden buildings in Ishinomaki city did not exceed three stories.
In this study, the lateral resistance of a building against tsunami impacts is considered as the sum of
lateral resistance for floors below the modelled maximum flow depth. Estimation of lateral resistance
for buildings should be taken with care as it was calculated for each floor. The total lateral resistance of
a building against seismic or wind loads would be the sum of lateral resistance for every floor where
maximum tsunami flow depth has reached. The highest estimated lateral resistance between seismic
and wind loads was then chosen as the maximum effective resistance, hence the assumed lateral
resistance design for each building. It should also be noted that the design lateral resistance may
decrease due to age and ground shaking from previous earthquakes. A previous study done by the Japan
Building Disaster Prevention Association (2012) reported 0.7 as the minimum reduction coefficient to
account for these effects. Therefore, a range of bearing wall resistance reduction coefficients (0.7, 0.8,
0.9 and 1.0) was introduced when calculating the lateral resistance of the building.

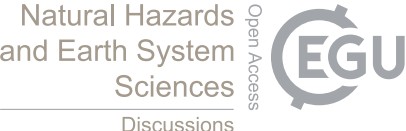
**Example 1**
Calculation example of required wall length for seismic load
One story with 60 m$^2$ of floor area, the required wall length = 60 m$^2$ × 15 cm/m$^2$ = 900 cm = 9 m
**Example 2**
Calculation example of required wall length for wind load
The first floor of two stories building,
Front: Required wall length = ①A (m$^2$) × 50 cm/m$^2$
Side: Required wall length = ②B (m$^2$) × cm/50 m$^2$
The second floor of two stories building
Front: Required wall length = ②A (m$^2$) × 50 cm/m$^2$
Side: Required wall length = ②B (m$^2$) × cm/50 m$^2$
The design wall length for wind load will be the summation of the maximum value at each floor.


| | |
|---|---|
| 15 cm/m$^2$ | One story building |
| 33 cm/m$^2$ | The first floor of two stories building |
| 21 cm/m$^2$ | The second floor of two stories |
| 50 cm/m$^2$ | The first floor of three stories building |
| 39 cm/m$^2$ | The second floor of three stories building |
| 24cm/m$^2$ | The third floor of three stories building |


**Fig. 7** Design coefficients for calculating corresponding necessary wall length against seismic load for
1-3 stories wooden houses.


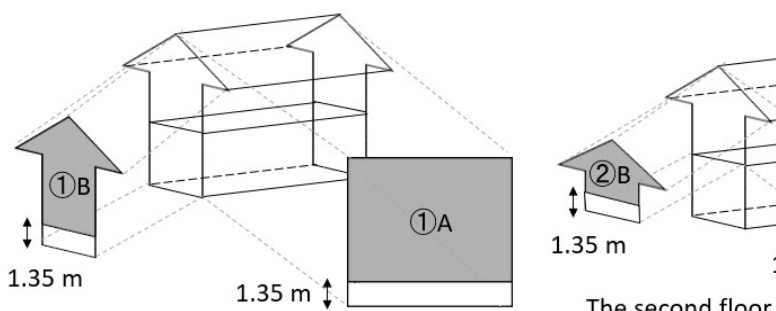

The first floor of two stories building

The second floor of two stories building


**Fig. 8** Calculation example of corresponding necessary wall length against wind load.

**2.5 Building damage replacement cost ratio**
Although financial loss is not the central focus of this paper, it is a good opportunity to present a
potential building damage replacement cost index for wooden buildings for future loss estimates. At
present, tsunami building damage costs are based on data obtained from insurance claims after tsunami
events. Loss estimates are, for the most part, based on analyses which are separate from the damage
assessments and they do not account for building conditions and tsunami hydrodynamics.
The building damage levels proposed by MLIT (**Fig. 3**) formed the basis of developing the replacement
cost index. Throughout this study, the focus has been on collapsed buildings (levels 5 and 6). This index
however will be representative of both collapsed and non-collapsed buildings. Collapsed buildings can
automatically be assigned as 100% loss as they are assumed to be irreparable. In general, construction
costs of two-storey wooden houses in Japan comprise two components – architectural works which
forms 70% of total costs and structural works which forms 30%. Costs of structural works can be further
broken down into non-structural components (roofs (20%) and walls (10%)) and structural components
(beams (20%), columns (15%) and footings (45%)) of the building. The averaged numbers of each
component were calculated based on actual data of several houses (MN Housing and Building
Laboratory, 2015, Cabinet Office of Japan, 2017, and Japan Wood-Products Information and Research
Center, 2019,).

**3.   Results and discussion**
**3.1 Accuracy of the proposed building damage assessment method**
The results of the proposed building damage assessment model were compared to field observations to
assess its performance (**Fig. 9**). Field observations are presented in the MLIT database and only
buildings with damage levels 5 and 6 (collapse conditions) were used for comparison. **Table 1** shows
an accuracy of modelled collapsed buildings and actual collapsed buildings from field observations
when only sliding mechanism was considered, and **Table 2** when both sliding and overturning
mechanisms were considered. Both tables have clearly illustrated that debris impact forces and
resistance reduction coefficients do not seem to have significantly influenced the collapse of buildings
in Ishinomaki. Damage analysis without debris weight input and building resistance reduction
coefficient showed a better match. This can be attributed to the fact that Ishinomaki city was not heavily
affected by floating debris for the reasons stated in **section 3.1.**

**Tables 3** and **4** highlight sliding mechanism alone is a poor explanation of collapse. In other words,
overturning is an important mechanism when analyzing building collapse. When using the proposed





method, the modelled results show a near 100% accuracy, as shown in **Table 4** and illustrated in **Fig.**
**9**.
**Table 1** Damage assessment accuracy (%): Washed away and destroyed buildings (damage levels 5
and 6) by considering only sliding as damage mechanism.

| Debris weight | Resistance reduction coefficient | | | |
|---|---|---|---|---|
| | 1 | 0.9 | 0.8 | 0.7 |
| 0 ton | 65.24 | 66.54 | 68.02 | 69.84 |
| 0.5 tons | 59.27 | 60.44 | 61.86 | 63.61 |
| 3 tons | 61.43 | 62.92 | 64.55 | 66.39 |
| 15 tons | 67.45 | 68.88 | 70.56 | 72.26 |
| 30 tons | 72.44 | 72.21 | 71.13 | 69.43 |
| 60 tons | 89.32 | 89.40 | 89.49 | 59.48 |


**Table 2** Damage assessment accuracy (%): Washed away and destroyed buildings (damage levels 5
and 6) by considering both damage mechanisms.

| Debris weight | Resistance reduction coefficient | | | |
|---|---|---|---|---|
| | 1 | 0.9 | 0.8 | 0.7 |
| 0 ton | 99.79 | 99.77 | 99.73 | 99.69 |
| 0.5 tons | 96.46 | 96.44 | 96.40 | 96.35 |
| 3 tons | 96.29 | 96.19 | 96.03 | 95.81 |
| 15 tons | 91.97 | 91.25 | 90.17 | 88.96 |
| 30 tons | 85.37 | 83.71 | 81.67 | 79.49 |
| 60 tons | 93.73 | 93.77 | 93.83 | 72.26 |


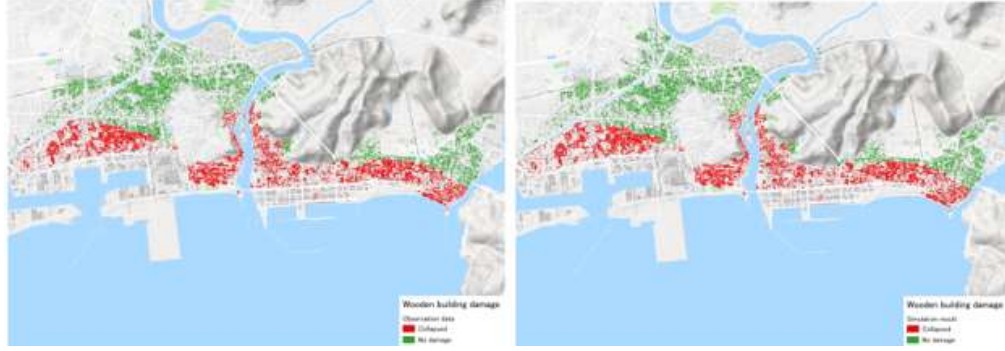


**Fig. 9** Distributions of collapsed and non-collapsed buildings from field observation (left) and the
proposed method (right)
**3.2 Comparison of minimum load values for the collapse of wooden buildings against field**
**observations and hydraulic experiments**
The average lateral resistance of a building in Ishinomaki, derived from 19, 000 wooden houses in this
study, is estimated to be about 42 kN, and the average hydrodynamic force is about 10 kN. These
findings are evaluated and compared to other findings in tsunami literature to understand the dominant
mechanism of building collapse. In a hydraulic experiment by Arikawa (2009), the flexural capacity of
a wooden wall was tested. A wooden wall (2.5 m high and 2.7 wide) supported by a steel frame was
placed in a water flume in a full-scale experiment. The wooden wall was found to be destroyed at a



tsunami flow depth of 2.5 m. The flexural capacity of the wooden wall was 10 kN/m², which is equivalent to 67.5 kN. Matsutomi and Harada (2010) measured tsunami flow depth at the front and back of buildings during their field survey. Based on the survey and estimated Froude number, they found that for wooden houses, the necessary lateral force required to cause moderate damage is 5.4 – 9.9 kN/m and for major damage, 9.7 – 17.6 kN/m. Therefore, the minimal lateral load required for wooden houses to be washed away is approximately 9.7 – 17.6 kN/m or 88 -176 kN, assuming that the width of the house is 5 – 10 m. This information further supports the consideration of overturning as a critical explanation for collapse mechanism.

### 3.3 Tsunami characteristics at the time of collapse and influence of flow characteristics on damage

Critical flow depth ($D_c$) and critical flow velocity ($V_c$) values are flow depths and velocities at the time of building collapse or rather, when buildings were considered collapsed when using the proposed damage model. In this study, a further assessment was made to derive maximum flow values and compare them to the critical values modelled for each building. In general, the critical values are lower than maximum values for both flow depth and velocity (**Figs. 10 & 11**). The maximum flow depth ($D_m$) is about four times higher than the critical flow depth and maximum flow velocity ($V_m$) is about two times higher than the critical flow velocity (**Table 3**). The implication is straightforward – building damage would be highly underestimated when using maximum flow characteristics as explanatory variables. It underscores one of the weaknesses of using traditional tsunami damage assessment methodologies.

It is also observed that flow depth and flow velocity contribute differently to total building damage. Critical flow depth and velocity for collapsed (damage levels 5 and 6) and non-collapsed buildings are plotted in **Fig. 12** and it appears that wooden buildings would almost always get washed away when critical flow velocity exceeds 2 m/s, regardless of the value of critical flow depth. This value may serve as a simple indicative criterion to assess building damage potential. This criterion when used together with developed tsunami maps or numerical flow simulation allows for some initial building damage assessment and quick estimations.

The influence of flow depth and flow velocity on building damage may also vary across space. The relationship between critical and maximum flow depth values are represented as ratios and the distribution of these ratios are plotted in a map (**Fig. 13 (Left)**). Similarly, the distribution of the ratio between critical and maximum flow velocities are plotted in a map (**Fig. 13 (Right)**). Flow velocity appears to be a more significant parameter of damage (as ratios are close to 1.00) in areas nearer to the shoreline where flow velocity is very high and tsunami induced force is mostly hydrodynamic. On the other hand, flow depth has a greater influence on damage in areas nearer to the inundation limit where pressure from the tsunami is mostly hydrostatic.





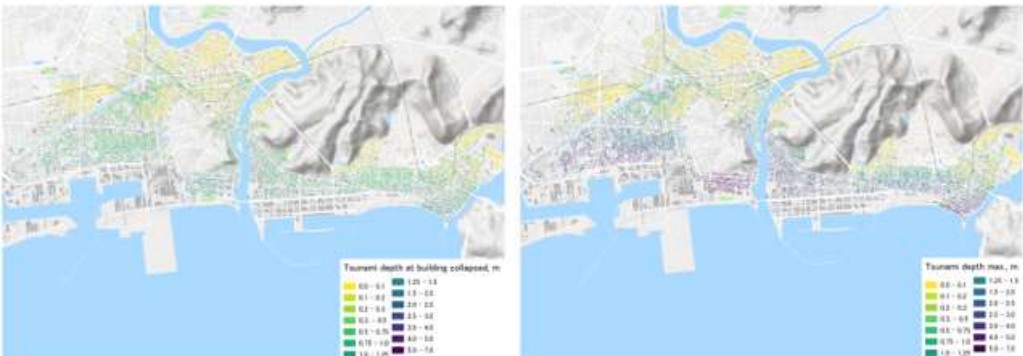


**Fig. 10** Distribution of the critical flow depth (left) and the maximum flow depth (right)

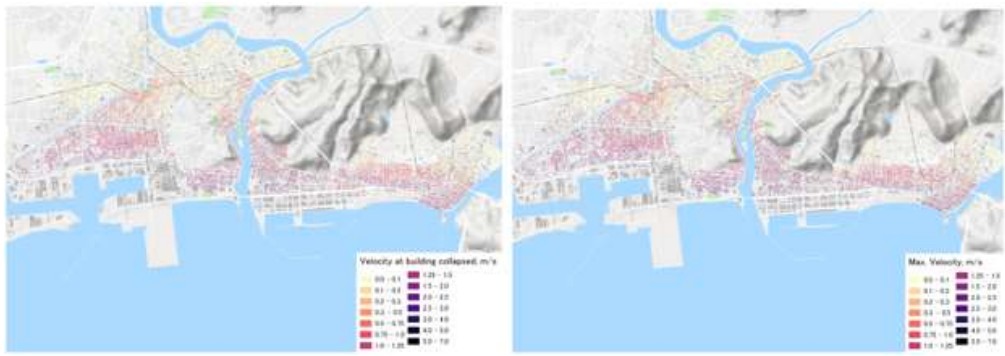


**Fig. 11** Distribution of the critical flow velocity (left) and the maximum flow velocity (right)

**Table 3** Flow depth and velocity ratios (washed away and destroyed buildings: damages levels 5 and
6).

| Damage conditions | $D_m / D_c$ | $V_m / V_c$ |
|---|---|---|
| Collapsed | 4.03 | 2.34 |
| Non-collapsed | 1.56 | 1.16 |


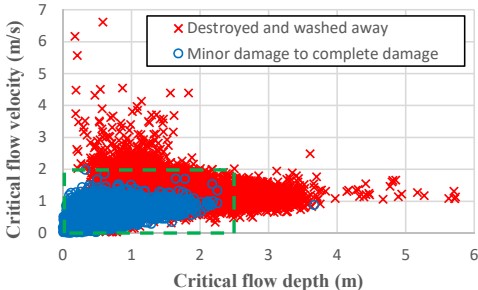


**Fig.12** Plotting of the critical flow depth and critical flow velocity




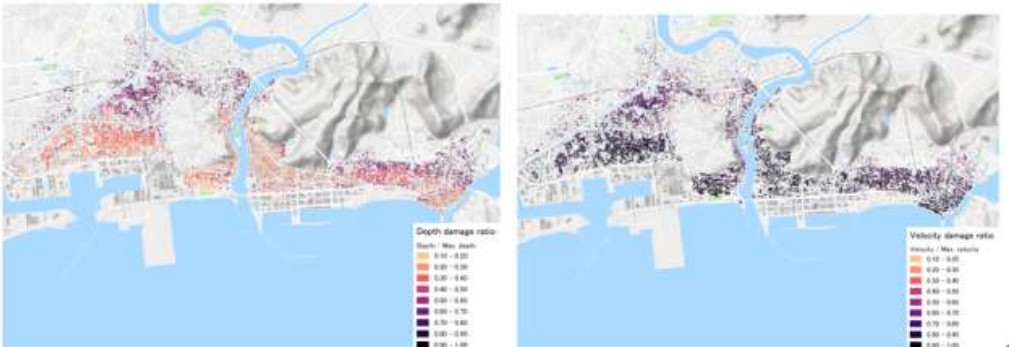


**Fig. 13** Distributions of ratios between the critical and the maximum values of the simulated flow
depth (left) and flow velocity (right). Higher ratios are found near inundation limit for the flow depth
whereas near shoreline for the flow velocity.


**3.4 Comparing results from fragility functions**

Building collapse in Ishinomaki City was recently modelled by Hasegawa et al. (2018), where they
developed fragility functions using the same building damage dataset (MLIT, 2012) and collapse
criteria. The fragility functions were developed by applying logistic regression (where damage states
follow a binomial distribution). The estimated damage probabilities are calculated as per equation (7).
Values of the maximum likelihood estimations are presented in **Table 4**.

$$p = \frac{1}{1 + \exp(-a_0 - a_i x_i - \cdots)} \qquad (7)$$

Where $p$ is a probability of collapse, $a_n$ is a regression constant and $x_n$ is an explanatory variable. In the
damage assessment of this study, a building is classified as collapsed when the probability of collapse
is higher than 50%.

**Table 4** The maximum likelihood estimates (Hasegawa et al., 2018)

|  | Estimate | Stand. Error | Z value | Pr (>\|z\|) |  |
|---|---|---|---|---|---|
| Constant term | -3.9250 | 0.0514 | -76.4360 | < 2e-16 | *** |
| RC building | -1.7970 | 0.0814 | -22.0870 | < 2e-16 | *** |
| Wooden building | 1.4120 | 0.0440 | 32.1180 | < 2e-16 | *** |
| Numbers of stories | -0.4242 | 0.0164 | -25.8550 | < 2e-16 | *** |
| Functions | 0.2272 | 0.0277 | 8.2050 | 2.31E-16 | *** |
| Flow depth | 1.0530 | 0.0060 | 174.1830 | < 2e-16 | *** |
| Building area | -0.0003 | 0.0000 | -7.1890 | 6.53E-13 | *** |

p value: *** > 0.001, ** >0.01, * > 0.05 and . > 0.1





Results from this study are compared to the fragility functions to determine how well building damage
can be identified when using either the proposed method or the fragility functions. The building damage
condition is reproduced using both methods and compared to actual observations as shown in **Fig. 14**.
The proposed method is able to correctly reproduce collapsed and non-collapsed buildings with 99.79%
accuracy, while the fragility functions are able to reproduce building damage conditions with 91.06%
accuracy, as summarized in **Table 5**. It can be observed the model based on fragility functions does not
perform as well when assessing building damage in the zone separating collapsed and non-collapsed
buildings.
It should be noted that building damage assessment with such accuracy can only be replicated because
of the strict construction design standards in Japan. How well the proposed method will perform in a
context outside of Japan will be largely dependent on local practices in the design and construction of
the buildings, the presence debris material and the age of the building (resistance reduction coefficients).
Additionally, flow-building interactions which yield lower damage states are not accounted for, so the
model may not perform as well for flow conditions which are less severe than the 2011 Great East Japan
tsunami.

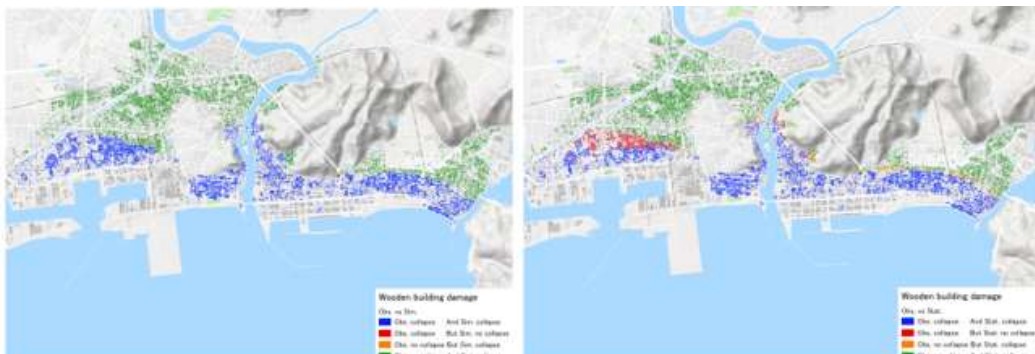


**Fig. 14** Reproduction of building damage condition (collapse or non-collapse): 1) Comparison between
the proposed method and field observation and 2) Fragility functions and field observation. Green:
Correct reproduction of collapsed buildings, Blue: Correct reproduction of non-collapsed buildings,
Red: Failed to reproduce collapsed buildings and Orange: Failed to reproduce non-collapsed buildings.









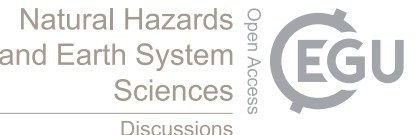

**Table 5** Building damage assessment accuracy of this proposed method and previously developed fragility functions compared to field observations. This table shows numbers of buildings for each condition and their accuracy percentages.

| | | Analytical method (this study) | |
| --- | --- | --- | --- |
| | | Collapsed | Non-collapsed |
| Field observation | Collapsed | 8,518 (45.22%) | 33 (0.18%) |
| | Non-collapsed | 7 (0.04%) | 10,277 (54.56%) |

| | | Fragility functions | |
| --- | --- | --- | --- |
| | | Collapsed | Non-collapsed |
| Field observation | Collapsed | 7,362 (39.09%) | 1,189 (6.31%) |
| | Non-collapsed | 519 (2.76%) | 9,765 (51.85%) |

## 3.5 Financial loss metrics

On account of these approximations of the construction cost, each building damage level defined by structural damage condition can be converted to replacement cost ratio as follows (see also **Table 6**).

Damage level 1: Minor damage (Replacement cost ratio = 18%)
Because of its damage description as "no significant structural or non-structural damage, possibly only minor flooding". A 25% architectural works is applied as the condition "Possible to be use immediately after minor floor and wall clean up".

Damage level 2: Moderate damage (Replacement cost ratio = 36%)
A damage ratio of 10% is assigned to roof and wall according to the damage description "Slight damages to non-structural components". A 50% architectural works is applied as the condition "Possible to be use after moderate reparation".

Damage level 3: Major damage (Replacement cost ratio = 54%)
A damage ratio of 25% is assigned to roof and wall according to the damage description "Heavy damages to some walls but no damages in columns". A 75% architectural works is applied as the condition "Possible to be use after major reparation".

Damage level 4: Complete damage (Replacement cost ratio = 76%)
A damage ratio of 50% is assigned to roof and wall and 25% to beam and column according to the damage description "Heavy damages to several walls and some columns". A 100% architectural works is applied as the condition "Possible to be use after a complete reparation and retrofitting".

Damage level 5: Collapsed (Replacement cost ratio = 100%)
A damage ratio of 75% is assigned to roof and wall and 50% to beam and column according to the damage description "Destructive damage to walls (more than half of wall density) and several columns (bend or destroyed). However, because a damage ratio of 100% is assigned to footing because of the



damage condition "Non-repairable or great cost for retrofitting", the final replacement cost ratio is set
to 100%.

Damage level 6: Washed away (Replacement cost ratio = 100%)
A damage ratio of 100% is assigned to all structural components according to the damage description
"Washed away, only foundation remained, total overturned" and damage condition "Non-repairable,
requires total reconstruction".

**Table 6** MLIT's damage level classification, description and condition (MLIT, 2012) and the
converted cost replacement ratios

| Damage level | Classification | Description | Condition | Replacement ratio |
|---|---|---|---|---|
| 1 | Minor damage | There is no significant structural or non-structural damage, possibly only minor flooding | Possible to be use immediately after minor floor and wall clean up | 18% |
| 2 | Moderate damage | Slight damages to non-structural components | Possible to be use after moderate reparation | 36% |
| 3 | Major damage | Heavy damages to some walls but no damages in columns | Possible to be use after major reparation | 54% |
| 4 | Complete damage | Heavy damages to several walls and some columns | Possible to be use after a complete reparation and retrofitting | 76% |
| 5 | Destroyed or collapsed | Destructive damage to walls (more than half of wall density) and several columns (bend or destroyed) | Loss of functionality (system collapse). Non-repairable or great cost for retrofitting | 100% |
| 6 | Washed away | Washed away, only foundation remained, total overturned | Non-repairable, requires total reconstruction | 100% |


**4. Conclusions**
This study presented a novel quantitative tsunami damage prediction approach, load-resistance analysis.
While previous empirical and experimental studies have vastly improved our understanding of building
response to tsunami impacts and extensively quantified building damage characteristics,
implementation of the resulting damage estimates for future tsunami scenarios is challenging; in
particular, when spatial differences such as construction standards and coastal morphology are
significant. Load-resistance analysis utilizes building design standards to estimate the resistance force
of each building, hence analytically estimate the potential for building damage (collapse) in a localized
context. One of the advantages of load-resistance analysis is it can be extended to other areas where
existing empirical data is sparse, and modified to assess building collapse (sliding or overturning
mechanism). This approach is complementary to published statistical tsunami damage fragility
functions as demonstrated in the case study of Ishinomaki City.
To date, building damage characteristics have been treated separately from financial losses which are
often of interest to policy makers and planners. This study is a first attempt to combine building damage
estimations and financial losses. Using the established classification of building damage by MLIT,



building construction costs were evaluated and pegged to each damage level as replacement cost ratios.
The proposed replacement cost index provide an approximate estimate of potential financial losses in
areas where pre-existing disaster-related insurance claim settlements are lacking.

### 4.1 Main findings

Additional key findings emerging from this study and are summarized below:
- Analytical estimation of the potential for building collapse was calculated using building design
standards and accounting for resistance reduction coefficients, as well as tsunami hydrodynamic
force considering different debris weights. The most general case (resistance reduction coefficient
of 1.0 and 0 ton debris weight) yields the highest accuracy in estimating building collapse in
Ishinomaki city.
- Sliding alone is an insufficient explanation for building collapse. It is also important to consider
overturning mechanism.
- This study has confirmed that the use of maximum values for flow depth and velocity might
underestimate damage. Damage is likely to occur before flow depth and velocity reach maximum
values. The present results suggest a flow velocity of 2 m/s or more would trigger collapse for a
typical Japanese 2 story residential wood building
- The ratio between critical flow velocity and maximum flow velocity might be a useful alternative
damage intensity measure but needs further investigation – particularly in the light of intermediate
damage levels.
- The proposed load-resistance analysis shows higher accuracy in assessing building collapse
compared to previously developed fragility functions in the same study area.
- Replacement cost ratio for each level of MLIT damage classification are approximately 18%, 36%,
54%, 76%, 100% and 100% for damage levels 1, 2, 3, 4, 5 and 6 respectively.

### 4.2 Future applications and limitations

The newly proposed method can be applied globally, only where building design standards and related
information are known and enforced. However, such detailed analyses require higher computational
cost and data storage. The proposed method may only work in countries where building design codes
are strictly followed as in the case of Japan and for events generating heavy levels of damage.
Additionally, the reliability of building damage predictions using this method is dependent on the
accuracy of the numerical model. This depends on the availability and quality of information regarding
the hazard, the dominant damage mode assumed in the analysis and/or reference dataset, the assumed
debris weight coefficient and the resistance reduction coefficient employed. In absence of such
information, building damage estimates are subjected to significant uncertainty. Therefore, the
application of this method is not to produce absolute figures for damage estimates, but to be a useful
guideline for planning purposes and an alternative study for comparison.

**Acknowledgments**
This research was funded by JSPS Grant-in-Aid for Young Scientists (B) "Applying developed fragility
functions for the Global Tsunami Model (GTM)" (Grant No. 16K16371), JSPS-NRCT Bilateral
Research grant, Tokio Marine & Nichido Fire Insurance Co., Ltd., Willis Research Network (WRN)
and the Radchadapisek Sompoch Endowment Fund (2019), Chulalongkorn University (762003-CC).

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
