# Peer review of "Load-resistance analysis: An alternative approach to tsunami damage assessment applied to the 2011 Great East Japan tsunami"

_Natural Hazards and Earth System Sciences, 2019_

## Referee Comment (RC1) · Ceren Ozer Sozdinler (Referee) · 19 Jun 2019

REVIEW OF MANUSCRIPT "LOAD-RESISTANCE ANALYSIS: AN ALTERNATIVE APPROACH TO TSUNAMI DAMAGE ASSESSMENT APPLIED TO THE 2011 GREAT EAST JAPAN TSUNAMI"

By Anawat Suppasri, Kwanchai Pakoksung, Ingrid Charvet, Constance Ting Chua, Noriyuki Takahashi, Teraphan Ornthammarath, Panon Latcharote, Natt Leelawat, and Fumihiko Imamura

This manuscript presents a new method for tsunami damage assessment for buildings in inundation area due to 2011 Great East Japan Tsunami. It is proposed as an alternative approach for the buildings located where fragility curves (previously developed by the main author and this team) do not exist. Therefore, it seems as a one-step further approach of previous studies on tsunami damage.

The manuscript has proper state of the art observational and numerical approach for tsunami modelling, damage force calculation and load-resistance analysis. The outcomes of this study would provide clear indications for tsunami hazard and risk assessment in Japan and for further studies that can be applied to other regions in the world.

The language of manuscript is so clear and very easy to understand. There is no need for proof-reading.

As a result of above, I believe that this paper would be a significant contribution for the tsunami research community.

I think the manuscript would be in its best version with minor revision and some further clarifications. I am listing my suggestions for revision and small typo correction below:

- Page 3 Line 119: "….examined in Latcherote et al (2016).
- Page 3 Line 124: "….. refers to the rotation of a building about its foundation…." Do you mean "around its foundation?"
- Page 4 Figure 1: It is better to write the definition of forces, i.e. W, R, etc in figure's caption.
- Page 5 Figure 3: Better to write (with the courtesy of ….) as a reference for the photos
- Page 5 Line 194: Is the Manning's roughness coefficients used as a spatial distribution depending on different type of buildings in study area or just constant values for a specific area? Please clarify.
- Page 5 Line 195: There is extra space between words "land" and "use"
- Page 5 Line 197: better to identify that "at the time of occurrence in 2011"
- Page 5 Line 198: better to use ";" or "and" instead of "," between sentences.
- Page 6 Figure 4: In the legend of figure, T.P. is not clear.
- Page 7 Figure 5: T.P. in the legends is not clear. Also, the color boxes in the legend were shifted. Better to reposition.
- Page 7 Figure 6: In the legend of figure, T.P. is not clear.
- Page 7 Line 228: I think it should be "through" instead of "though"
- Page 7 Line 237: Better to write a short explanation about why you used CD=1.5
- Page 7 Line 238: What is the reference for using dt=0.7sec for wooden wall? Please specify.
- Page 8 Line 272: How do you assume 3.5m,2.7m,2.1m for height of buildings in each floor? Please give reference or at least make a short explanation. Because these values are so specific.

- Page 9 Figure 7: What is the reference for these design coefficients? Also it is confusing to see A, B without having any prior explanation. We understand their meaning only after seeing Figure 8. I think better to change the order of these figures. Besides, cm/50 m2 is confusing.
- Page 10 Line 340: I think they should be **"Table 1 and 2"** instead of "Table 3 and 4"
- Page 11 Line 342: I think it should be **"Table 2"** instead of "Table 4".
- Page 12 Line 365: "…for major damage is 9.7-17.6 kN/m…."
- Page 13 Figure 10: This figure needs further explanation, especially in the caption. Are they observed or calculated values; it is not clear. Also, better to reposition color boxes in color legend (shifted).
- Page 14 Figure 13: The label "Depth/max/ depth" in color legend is confusing. Maybe better to write "critical vs max depth ratio" instead.
- Page 14 Table 4: Is there any mistake in the last column? Because in the p values footnote at the bottom there is explanation for ** and * but all values in last column are ***.
- Page 14 Figure 14: (i) I think there is a mistake in color legend and caption. Explanation of Green in legend should be "Obs. No collapse and Stat. no collapse". (ii) In figure caption it is better to write "(left)" and "(right)" instead of 1) and 2). (iii) in the caption it should be "Blue: Correct reproduction of collapsed buildings" instead of Green:…. and "Green: Correct reproduction of non-collapsed buildings" instead of Blue:… (iv) it is better to write a title on each plot, i.e. "proposed method" for the left and "fragility curves" for the right one.
- Page 16 Section 3.5: I think this section is a bit confusing in total. First of all it is not clear how you assign 25%, 50%, 75% damage ratios. Then conversion of replacement ratios in next page on Table 6 is not clear as well. Please clarify this calculation.
- Page 17 Line 517: Related with the previous suggestion, this part is not clear "to combine building damage estimations and financial losses". Further explanation is needed.
- Page 18 Line 522: Please delete "and"
- Page 18 Section 4.2: I think this proposed method needs a name. Like "fragility curves" method or else, it would be good to give a name to this new proposed method for convenience in further studies and references. Also, if applicable, I think it is better to clearly remark that this proposed method can be used for wooden buildings located along other coastal regions of Japan. It would be good to specify this method would be applicable for other regions in Japan.

---

## Author Comment (AC1) · 21 Jun 2019

Please see our replies and corrections in detail in the supplement file.

Please also note the supplement to this comment:
https://www.nat-hazards-earth-syst-sci-discuss.net/nhess-2019-71/nhess-2019-71-AC1-supplement.pdf
* * *
[Figure]

[Figure]

**Fig. 1.** Fig 4

Max. Water level (m)

- 0
- 2
- 4
- 6
- 8
- 10
- 12

**Fig. 2.** Fig 5a

Max. Water velocity (m/s)

| | | | |
|---|---|---|---|
| ☐ 0 | ☐ 4 | ☐ 8 | ■ 12 |
| ■ 2 | ☐ 6 | ■ 10 | |

**Fig. 3.** Fig 5b

Tsunami depth at
building collapsed, m

| | |
|---|---|
| 0.0 – 0.1 | 1.25 – 1.5 |
| 0.1 – 0.2 | 1.5 – 2.0 |
| 0.2 – 0.3 | 2.0 – 2.5 |
| 0.3. – 0.5 | 2.5 – 3.0 |
| 0.5 – 0.75 | 3.0 – 4.0 |
| 0.75 – 1.0 | 4.0 – 5.0 |
| 1.0 – 1.25 | 5.0 – 7.0 |

**Fig. 4.** Fig 10a

Tsunami depth max., m

| | | | |
|---|---|---|---|
| ▨ | 0.0 − 0.1 | ▨ | 1.25 − 1.5 |
| ▨ | 0.1 − 0.2 | ▨ | 1.5 − 2.0 |
| ▨ | 0.2 − 0.3 | ▨ | 2.0 − 2.5 |
| ▨ | 0.3. − 0.5 | ▨ | 2.5 − 3.0 |
| ▨ | 0.5 − 0.75 | ▨ | 3.0 − 4.0 |
| ▨ | 0.75 − 1.0 | ▨ | 4.0 − 5.0 |
| ▨ | 1.0 − 1.25 | ▨ | 5.0 − 7.0 |

**Fig. 5.** Fig 10b

**Tsunami velocity at
building collapsed, m/s**

| | |
|---|---|
| ▨ 0.0 − 0.1 | ▨ 1.25 − 1.5 |
| ▨ 0.1 − 0.2 | ▨ 1.5 − 2.0 |
| ▨ 0.2 − 0.3 | ▨ 2.0 − 2.5 |
| ▨ 0.3. − 0.5 | ▨ 2.5 − 3.0 |
| ▨ 0.5 − 0.75 | ▨ 3.0 − 4.0 |
| ▨ 0.75 − 1.0 | ▨ 4.0 − 5.0 |
| ▨ 1.0 − 1.25 | ▨ 5.0 − 7.0 |

**Fig. 6.** Fig 11a

[Figure]

Max. Velocity, m/s

| | |
|---|---|
| 0.0 − 0.1 | 1.25 − 1.5 |
| 0.1 − 0.2 | 1.5 − 2.0 |
| 0.2 − 0.3 | 2.0 − 2.5 |
| 0.3. − 0.5 | 2.5 − 3.0 |
| 0.5 − 0.75 | 3.0 − 4.0 |
| 0.75 − 1.0 | 4.0 − 5.0 |
| 1.0 − 1.25 | 5.0 − 7.0 |

**Fig. 7.** Fig 11b

Depth damage ratio

Critical vs max. depth ratio

- 0.10 − 0.20
- 0.20 − 0.30
- 0.30 − 0.40
- 0.40 − 0.50
- 0.50 − 0.60
- 0.60 − 0.70
- 0.70 − 0.80
- 0.80 − 0.90
- 0.90 − 1.00

**Fig. 8.** Fig 13a

Velocity damage ratio

Critical vs max. velocity ratio

- 0.10 − 0.20
- 0.20 − 0.30
- 0.30 − 0.40
- 0.40 − 0.50
- 0.50 − 0.60
- 0.60 − 0.70
- 0.70 − 0.80
- 0.80 − 0.90
- 0.90 − 1.00

**Fig. 9.** Fig 13b

Wooden building damage

Stat. vs Sim.

- ■ Stat. collapse     And Sim. collapse
- ■ Stat. collapse     But Sim. no collapse
- ■ Stat. no collapse But Sim. collapse
- ■ Stat. no collapse And Sim. no collapse

Load-resistance analysis
(The proposed method)

**Fig. 10.** Fig 14a

Wooden building damage

Stat. vs Sim.

■ Stat. collapse      And Sim. collapse
■ Stat. collapse      But Sim. no collapse
■ Stat. no collapse But Sim. collapse
■ Stat. no collapse And Sim. no collapse

Fragility functions

**Fig. 11.** Fig 14b

**Supplement:**

We highly appreciate the time spent for the review comments from the reviewer especially those minor corrections (our typo errors) and pointed out many points that clarifications are needed. We are happy that the reviewer is happy and highly evaluated our manuscript. Please find our responses and corrections as shown below.

| Reviewer comments | Our answers | Corrected manuscript |
|---|---|---|
| - Page 3 Line 119: "….examined in Latcherote et al (2016). | Corrected | … examined in Latcharote et al. (2016). |
| - Page 3 Line 124: "….. refers to the rotation of a building about its foundation…." Do you mean "around its foundation?" | Corrected | …a building around its foundation… |
| - Page 4 Figure 1: It is better to write the definition of forces, i.e. W, R, etc in figure's caption. | Explanations are added in the caption | The forces denoted are as follows, $F_h$ = hydrodynamic force, $F_d$ = debris impact force, $R$ = lateral resistance, $W$ = building weight and $B$ = buoyancy force. |
| - Page 5 Figure 3: Better to write (with the courtesy of ….) as a reference for the photos | Explanations are added in the caption | …(courtesy of MLIT, 2012). |
| - Page 5 Line 194: Is the Manning's roughness coefficients used as a spatial distribution depending on different type of buildings in study area or just constant values for a specific area? Please clarify. | We used a constant value of Manning coefficient in regions 1-5. For region 6, we used specific value depending on land use and building density. | A constant value of Manning coefficient was applied to all computational grids except at the finest resolution (Region 6)… |
| - Page 5 Line 195: There is extra space between words "land" and "use" | The space is added. | …according to land use types… |
| - Page 5 Line 197: better to identify that "at the time of occurrence in 2011" | Explanations are added | …tsunami occurrence in 2011 and simulation… |
| - Page 5 Line 198: better to use ";" or "and" instead of "," between sentences. | Corrected | …deformation and the fault… |
| - Page 6 Figure 4: In the legend of figure, T.P. is not clear. | Corrected. Explanations are added in the caption. | Please see the corrected Fig. 4 Projection of bathymetry and topography data is the Japanese Geodetic Datum 2000 and the Tokyo Peil (T.P.) datum. |
| - Page 7 Figure 5: T.P. in the legends is not clear. Also, the color boxes in the legend were shifted. Better to reposition. | Corrected | Please see the corrected Fig. 5 |
| - Page 7 Figure 6: In the legend of figure, T.P. is not clear. | Corrected | Please see the corrected Fig. 6 |
| - Page 7 Line 228: I think it should be "through" instead of "though" | Corrected | …through drag formula… |
| - Page 7 Line 237: Better to write a short explanation about why you used CD=1.5 | Explanations are added | ($C_D$ = 1.5 as an average value from 1.25 to 2.00 depending on the width to depth ratio, FEMA, 2003), |
| - Page 7 Line 238: What is the | From FEMA | (= 0.7 sec for wooden wall, FEMA, 2003) |

| | | |
|---|---|---|
| reference for using dt=0.7sec for wooden wall? Please specify. | | |
| - Page 8 Line 272: How do you assume 3.5m,2.7m,2.1m for height of buildings in each floor? Please give reference or at least make a short explanation. Because these values are so specific. | They are average floor heights of wooden houses from an interview with local housing construction company. | …an anonymous interview was conducted with a local housing construction company. The estimates provided for the heights of the first, second and third floors of an average wooden housing were 3.5 m, 2.7 m and 2.1 m respectively, which were then used as the average values for the purpose of this study. |
| - Page 9 Figure 7: What is the reference for these design coefficients? Also it is confusing to see A, B without having any prior explanation. We understand their meaning only after seeing Figure 8. I think better to change the order of these figures. Besides, cm/50 m2 is confusing. | These coefficients are also from MLIT (2018). We have also moved Fig. 8 after Example 2 to avoid such confusion. Cm/50 m2 is a wrong typing. | Please see Fig. 7 and Example 2 |
| - Page 10 Line 340: I think they should be "Table 1 and 2" instead of "Table 3 and 4" | Corrected | Tables 1 and 2 highlight… |
| - Page 11 Line 342: I think it should be "Table 2" instead of "Table 4". | Corrected | …Table 2 and illustrated in Fig. 9. |
| - Page 12 Line 365: "…for major damage is 9.7-17.6 kN/m…." | Corrected | …major damage is 9.7 – 17.6 kN/m…. |
| - Page 13 Figure 10: This figure needs further explanation, especially in the caption. Are they observed or calculated values; it is not clear. Also, better to reposition color boxes in color legend (shifted). | Corrected. Explanations are added in the caption. Color boxes are also correct. | Please see Fig. 10. …the simulated critical flow depth (left) and the simulated maximum flow depth… |
| - Page 14 Figure 13: The label "Depth/max/ depth" in color legend is confusing. Maybe better to write "critical vs max depth ratio" instead. | Corrected | Please see Fig. 13. |
| - Page 14 Table 4: Is there any mistake in the last column? Because in the p values footnote at the bottom there is explanation for ** and * but all values in last column are ***. | We agreed with the reviewer that as there is only one type of p value, we have corrected accordingly. | p value: * < 0.001 |
| - Page 14 Figure 14: (i) I think there is a mistake in color legend and caption. Explanation of Green in legend should be "Obs. No collapse and Stat. no collapse". (ii) In figure caption it is better to write "(left)" and "(right)" instead of 1) and 2). (iii) in the caption it | We are sorry for our mistake both in the figures and captions. All is corrected now. We also added a text box in each figure to clearly mention that this is a comparison of our proposed method and the original fragility functions. | Please see Fig. 14. |

| | | |
|---|---|---|
| should be "Blue: Correct reproduction of collapsed buildings" instead of Green:…. and "Green: Correct reproduction of non-collapsed buildings" instead of Blue:… (iv) it is better to write a title on each plot, i.e. "proposed method" for the left and "fragility curves" for the right one. | | |
| - Page 16 Section 3.5: I think this section is a bit confusing in total. First of all it is not clear how you assign 25%, 50%, 75% damage ratios. Then conversion of replacement ratios in next page on Table 6 is not clear as well. Please clarify this calculation. | The 25%, 50%, 75% damage ratios were interpreted from MLIT's damage definition. For this, Table 6 (Former Table 7) is modified. Table 7 is newly added to summarize the assigned ratio to each structural component. In addition, calculation example of the replacement cost ratio for each damage level is also added. | Please see the revised section 3.5. |
| - Page 17 Line 517: Related with the previous suggestion, this part is not clear "to combine building damage estimations and financial losses". Further explanation is needed. | We have modified this sentence. | …first attempt to propose both building damage estimations and financial losses. |
| - Page 18 Line 522: Please delete "and" | Corrected | |
| - Page 18 Section 4.2: I think this proposed method needs a name. Like "fragility curves" method or else, it would be good to give a name to this new proposed method for convenience in further studies and references. Also, if applicable, I think it is better to clearly remark that this proposed method can be used for wooden buildings located along other coastal regions of Japan. It would be good to specify this method would be applicable for other regions in Japan. | We have modified this sentence. | The newly proposed load-resistance analytical method can be applied to other coastal regions of Japan and globally,… |

---

## Referee Comment (RC2) · Ceren Ozer Sozdinler (Referee) · 25 Jun 2019

Thanks the author to take into account my comments and review the manuscript accordingly. The revisions and clarifications in the new version are fine for me. I believe the manuscript in this current version is clear and appropriate to be published.

---

## Referee Comment (RC3) · Anonymous Referee #2 · 8 Jul 2019

Review of the manuscript: nhess-2019-71 Title: Load-resistance analysis: An alternative approach to tsunami damage assessment applied to the 2011 Great East Japan tsunami Authors: Anawat Suppasri, Kwanchai Pakoksung, Ingrid Charvet, Constance Ting Chua, Noriyuki Takahashi, Teraphan Ornthammarath, Panon Latcharote, Natt Leelawat and Fumihiko Imamura.

The manuscript addresses the fragility functions of coastal buildings under tsunami demand, with application to the 2011 Japan event. The MS is well-written and benefits from high-quality presentation and figures worth to be published in NHESS journal.

The authors concluded that they are proposing a novel approach allowing the assessment of tsunami damage for buildings in regions where fragility functions are not available. I agree with the utmost importance to develop alternative approaches for the tsunami damage assessment for coastal areas where there is a lack of recent tsunami damage data. However, I find that the present work overlooks the significant progress made in developing the tsunami fragility functions and doesn't take in consideration the recently published works, which makes the claimed novelty of the presented approach questionable.

The methods for deriving tsunami fragility functions can be broadly classified into four categories: (1) empirical methods based on statistical analysis of observed post-tsunami damage data (Peiris 2006; Dias et al. 2009; Suppasri et al. 2015); (2) hybrid techniques that combine damage data from remote sensing and hazard mapping (numerical simulation of tsunami inundation) (e.g., Koshimura et al. 2009a, b; Suppasri et al. 2011); (3) heuristic fragility functions based on expert opinion (e.g., FEMA 2013); and (4) analytical fragility functions based on structural modeling and response simulations (Macabuag et al. 2014; Nanayakkara and Dias 2016; Attary et al. 2017).

These classifications must appear somewhere in the MS to highlight that analytical-based approach for deriving tsunami fragility functions exists in the literature well before this submission. Therefore, the novelty of this work must be discussed in light of the above-mentioned works.

---

## Author Comment (AC3) · 18 Jul 2019

Thank you for your confirmation of our revision. We are happy that our revision is satisfied.

---

## Author Response (AR1)

**Dear Editor,**

Thank you for your time and sending us your decision. We have made corrections to both reviewers as shown below. I am sorry that I forgot to add author response table for reviewer no. 2 in the interactive discussion page. Corrections made based on suggestions from reviewer no. 1 are shown in red and reviewer no.2 are shown in blue.

**Reply to reviewer no. 1**
We highly appreciate the time spent for the review comments from the reviewer especially those minor corrections (our typo errors) and pointed out many points that clarifications are needed. We are happy that the reviewer is happy and highly evaluated our manuscript. Please find our responses and corrections as shown below.

| Reviewer comments | Our answers | Corrected manuscript |
|---|---|---|
| - Page 3 Line 119: "….examined in Latcherote et al (2016). | Corrected | … examined in Latcharote et al. (2016). |
| - Page 3 Line 124: "….. refers to the rotation of a building about its foundation…." Do you mean "around its foundation?" | Corrected | …a building around its foundation… |
| - Page 4 Figure 1: It is better to write the definition of forces, i.e. W, R, etc in figure's caption. | Explanations are added in the caption | The forces denoted are as follows, $F_h$ = hydrodynamic force, $F_d$ = debris impact force, $R$ = lateral resistance, $W$ = building weight and $B$ = buoyancy force. |
| - Page 5 Figure 3: Better to write (with the courtesy of ….) as a reference for the photos | Explanations are added in the caption | …(courtesy of MLIT, 2012). |
| - Page 5 Line 194: Is the Manning's roughness coefficients used as a spatial distribution depending on different type of buildings in study area or just constant values for a specific area? Please clarify. | We used a constant value of Manning coefficient in regions 1-5. For region 6, we used specific value depending on land use and building density. | A constant value of Manning coefficient was applied to all computational grids except at the finest resolution (Region 6)… |
| - Page 5 Line 195: There is extra space between words "land" and "use" | The space is added. | …according to land use types… |
| - Page 5 Line 197: better to identify that "at the time of occurrence in 2011" | Explanations are added | …tsunami occurrence in 2011 and simulation… |
| - Page 5 Line 198: better to use ";" or "and" instead of "," between sentences. | Corrected | …deformation and the fault… |
| - Page 6 Figure 4: In the legend of figure, T.P. is not clear. | Corrected. Explanations are added in the caption. | Please see the corrected Fig. 4 Projection of bathymetry and topography data is the Japanese Geodetic Datum 2000 and the Tokyo Peil (T.P.) datum. |
| - Page 7 Figure 5: T.P. in the legends is not clear. Also, the color boxes in the legend were shifted. Better to reposition. | Corrected | Please see the corrected Fig. 5 |
| - Page 7 Figure 6: In the legend of | Corrected | Please see the corrected Fig. 6 |

| figure, T.P. is not clear. | | |
|---|---|---|
| - Page 7 Line 228: I think it should be "through" instead of "though" | Corrected | …through drag formula… |
| - Page 7 Line 237: Better to write a short explanation about why you used CD=1.5 | Explanations are added | ($C_D$ = 1.5 as an average value from 1.25 to 2.00 depending on the width to depth ratio, FEMA, 2003), |
| - Page 7 Line 238: What is the reference for using dt=0.7sec for wooden wall? Please specify. | From FEMA | (= 0.7 sec for wooden wall, FEMA, 2003) |
| - Page 8 Line 272: How do you assume 3.5m,2.7m,2.1m for height of buildings in each floor? Please give reference or at least make a short explanation. Because these values are so specific. | They are average floor heights of wooden houses from an interview with local housing construction company. | …an anonymous interview was conducted with a local housing construction company. The estimates provided for the heights of the first, second and third floors of an average wooden housing were 3.5 m, 2.7 m and 2.1 m respectively, which were then used as the average values for the purpose of this study. |
| - Page 9 Figure 7: What is the reference for these design coefficients? Also it is confusing to see A, B without having any prior explanation. We understand their meaning only after seeing Figure 8. I think better to change the order of these figures. Besides, cm/50 m2 is confusing. | These coefficients are also from MLIT (2018). We have also moved Fig. 8 after Example 2 to avoid such confusion. Cm/50 m2 is a wrong typing. | Please see Fig. 7 and Example 2 |
| - Page 10 Line 340: I think they should be "Table 1 and 2" instead of "Table 3 and 4" | Corrected | Tables 1 and 2 highlight… |
| - Page 11 Line 342: I think it should be "Table 2" instead of "Table 4". | Corrected | …Table 2 and illustrated in Fig. 9. |
| - Page 12 Line 365: "…for major damage is 9.7-17.6 kN/m…." | Corrected | …major damage is 9.7 – 17.6 kN/m…. |
| - Page 13 Figure 10: This figure needs further explanation, especially in the caption. Are they observed or calculated values; it is not clear. Also, better to reposition color boxes in color legend (shifted). | Corrected. Explanations are added in the caption. Color boxes are also correct. | Please see Fig. 10.
…the simulated critical flow depth (left) and the simulated maximum flow depth… |
| - Page 14 Figure 13: The label "Depth/max/ depth" in color legend is confusing. Maybe better to write "critical vs max depth ratio" instead. | Corrected | Please see Fig. 13. |
| - Page 14 Table 4: Is there any mistake in the last column? Because in the p values footnote at the bottom there is explanation for ** and * but all values in last column are ***. | We agreed with the reviewer that as there is only one type of p value, we have corrected accordingly. | p value: * < 0.001 |

| | | |
|---|---|---|
| - Page 14 Figure 14: (i) I think there is a mistake in color legend and caption. Explanation of Green in legend should be "Obs. No collapse and Stat. no collapse". (ii) In figure caption it is better to write "(left)" and "(right)" instead of 1) and 2). (iii) in the caption it should be "Blue: Correct reproduction of collapsed buildings" instead of Green:…. and "Green: Correct reproduction of non-collapsed buildings" instead of Blue:… (iv) it is better to write a title on each plot, i.e. "proposed method" for the left and "fragility curves" for the right one. | We are sorry for our mistake both in the figures and captions. All is corrected now. We also added a text box in each figure to clearly mention that this is a comparison of our proposed method and the original fragility functions. | Please see Fig. 14. |
| - Page 16 Section 3.5: I think this section is a bit confusing in total. First of all it is not clear how you assign 25%, 50%, 75% damage ratios. Then conversion of replacement ratios in next page on Table 6 is not clear as well. Please clarify this calculation. | The 25%, 50%, 75% damage ratios were interpreted from MLIT's damage definition. For this, Table 6 (Former Table 7) is modified. Table 7 is newly added to summarize the assigned ratio to each structural component. In addition, calculation example of the replacement cost ratio for each damage level is also added. | Please see the revised section 3.5. |
| - Page 17 Line 517: Related with the previous suggestion, this part is not clear "to combine building damage estimations and financial losses". Further explanation is needed. | We have modified this sentence. | …first attempt to propose both building damage estimations and financial losses. |
| - Page 18 Line 522: Please delete "and" | Corrected | |
| - Page 18 Section 4.2: I think this proposed method needs a name. Like "fragility curves" method or else, it would be good to give a name to this new proposed method for convenience in further studies and references. Also, if applicable, I think it is better to clearly remark that this proposed method can be used for wooden buildings located along other coastal regions of Japan. It would be good to specify this method would be applicable for other regions in Japan. | We have modified this sentence. | The newly proposed load-resistance analytical method can be applied to other coastal regions of Japan and globally,… |

**Reply to reviewer no. 2**

Thank you so much for your suggestion in summarizing previous researches. We have added more explanations in section 1 (Introduction) as shown below.

| Reviewer comments | Our answers | Corrected manuscript |
|---|---|---|
| - The manuscript addresses the fragility functions of coastal buildings under tsunami demand, with application to the 2011 Japan event. The MS is well-written and benefits from high-quality presentation and figures worth to be published in NHESS journal. The authors concluded that they are proposing a novel approach allowing the assessment of tsunami damage for buildings in regions where fragility functions are not available. I agree with the utmost importance to develop alternative approaches for the tsunami damage assessment for coastal areas where there is a lack of recent tsunami damage data. | Thank you for your time reviewing our paper and we are happy that our work is satisfied and highly evaluated. Please see our corrections to your suggestions in the second comment. | |
| - However, I find that the present work overlooks the significant progress made in developing the tsunami fragility functions and doesn't take in consideration the recently published works, which makes the claimed novelty of the presented approach questionable. The methods for deriving tsunami fragility functions can be broadly classified into four categories: (1) empirical methods based on statistical analysis of observed post tsunami damage data (Peiris 2006; Dias et al. 2009; Suppasri et al. 2015); (2) hybrid techniques that combine damage data from remote sensing and hazard mapping (numerical simulation of tsunami inundation) (e.g., Koshimura et al. 2009a, b; Suppasri et al. 2011); (3) heuristic fragility functions based on expert opinion (e.g., FEMA 2013); and | We have added more explanations of the four items as shown in blue. | Tsunami fragility functions are modelled using tsunami flow characteristics and building damage information. In general, the methods for deriving tsunami fragility functions can be classified into four categories. (1) Empirical methods based on statistical analysis of observed post tsunami damage data (e.g., Peiris, 2006, Reese et al., 2007, Dias et al. 2009, Valencia et al., 2011, Suppasri et al. 2015 and Triantafyllou et al., 2018). In a field survey, maximum flow depth measured from tsunami water traces are typically used as explanatory variables of damage. Building damage data is obtained from on-site observations. (2) Hybrid techniques that combine tsunami hazard mapping (numerical simulation of tsunami inundation such as maximum flow |

| | | |
|---|---|---|
| (4) analytical fragility functions based on structural modeling and response simulations (Macabuag et al. 2014; Nanayakkara and Dias 2016; Attary et al. 2017). These classifications must appear somewhere in the MS to highlight that analytical based approach for deriving tsunami fragility functions exists in the literature well before this submission. Therefore, the novelty of this work must be discussed in light of the above-mentioned works. | | depth, maximum flow velocity and maximum hydrodynamic force) with interpreted building damage data from remote sensing and (e.g., Koshimura et al. 2009, Omira et al., 2010 and Suppasri et al. 2011) or other damage data set such as damaged marine vessels (Suppasri et al., 2014), damaged bridges (Shoji and Nakamura, 2017) as well as aquaculture rafts and eelgrass (Suppasri et al., 2018).

(3) Heuristic fragility functions based on expert opinion such as HAZUS (FEMA 2013) and Papathoma Tsunami Vulnerability Assessment (PTVA) (Dall'Osso et al., 2016).

(4) Analytical fragility functions based on structural modelling and response simulations (e.g. Macabuag et al. 2014, Nanayakkara and Dias 2016 and Attary et al. 2017). |

---

## Author Response (AR2)

**nhess-2019-71**

**Load-resistance analysis: An alternative approach to tsunami damage assessment applied to the 2011 Great East Japan tsunami**

Dear Editor,

Thank you for your final comment with your decision to accept our manuscript. We would like to answer your question that our proposed method is not that mentioned four methods in introduction section. In other word, our proposed method is an alternative approach that we may not have to rely on fragility functions. We have added more clarification about this in introduction section in final version of our manuscript.

Best regards,
Anawat Suppasri, on behalf of all co-authors